



# North Atlantic marine organic aerosol characterized by novel offline thermal desorption mass spectrometry approach: polysaccharides, recalcitrant material, secondary organics

Michael J. Lawler[1], Savannah L. Lewis[2], Lynn M. Russell[2], Patricia K. Quinn[3], Timothy S. Bates[4,3], Derek J. Coffman[3], Lucia M. Upchurch[4,3], and Eric S. Saltzman[1]

[1]Department of Chemistry, Department of Earth System Science, University of California, Irvine
[2]Scripps Institution of Oceanography, University of California, San Diego
[3]Pacific Marine Environmental Laboratory, National Oceanic and Atmospheric Administration, Seattle, WA
[4]Joint Institute for the Study of the Atmosphere and Ocean, University of Washington, Seattle, WA

*Correspondence to*: Michael J. Lawler (mlawler@uci.edu)

**Abstract.** The composition of organic compounds in marine aerosols and the relative contributions of primary and secondary organic compounds remain uncertain. We report results from a novel approach to characterize and quantify organic components of the marine aerosol. Size-segregated discrete aerosol filter samples were collected at sea in the North Atlantic from both ambient aerosol and artificially generated primary sea spray over four cruises timed to capture the seasonal phytoplankton bloom dynamics. Samples were analyzed by Fourier transform infrared spectroscopy (FTIR), extracted into water, and analyzed by offline thermal desorption chemical ionization mass spectrometry (TDCIMS) and ion chromatography (IC). A positive matrix factorization (PMF) analysis identified several characteristic aerosol components in the TDCIMS mass spectra. Among these is a "polysaccharide factor" representing about 10-30% of the submicron organic aerosol mass. An unquantified "recalcitrant factor" of highly thermally stable organics showed significant correlation with FTIR-measured alcohol groups, consistently the main organic functional group associated with sea spray aerosol. We hypothesize that this factor represents recalcitrant dissolved organic matter in seawater. The recalcitrant factor showed little seasonal variability in its contribution to primary marine aerosol. The relative contribution of polysaccharides was highest in late spring and summer in the smallest particle size fraction characterized (<180 nm).

## 1 Background

Marine aerosols are important to the climate system because of their direct interactions with incoming solar radiation and their indirect effect on cloud radiative properties. Marine aerosols are a complex and variable mixture of locally generated material (emissions of particles and gases from the sea surface) and transported material from the continents (pollutants, biomass burning, desert dust). It is well known that marine aerosol is formed through the physical ejection of surface ocean material as sea spray (primary aerosol formation), and through gas to particle conversion of precursor gases, such as dimethylsulfide,



which lead to secondary aerosol formation (Clarke et al., 1998; O'Dowd and de Leeuw, 2007; Prospero, 2002; Sanchez et al., 2018). In particular, organic matter makes up a significant fraction of the submicron marine aerosol, but the chemical composition and origin of this material is not well characterized. These organics may arise from bubble bursting, emissions of

biogenic aerosol precursors like isoprene and monoterpenes, or from oxygenated aerosol-forming volatile organic compounds perhaps arising from photochemistry occurring on the sea surface (Bernard et al., 2016; Mungall et al., 2017).

There is considerable evidence showing a significant contribution from primary marine organics that are ejected as sea spray from bubble bursting at the ocean surface (Bates et al., 2012; Frossard et al., 2014; Russell et al., 2010). The origin of these organics is the pool of dissolved organic matter (DOM) and particulate organic matter (POM) in the surface ocean, but these

materials are not conveyed to the primary aerosol in the same proportions as are present in the surface ocean. Smaller (submicron) sea spray particles are enriched in surface-active organics with respect to the bulk seawater dissolved compounds because these "film drops" arise from the thin, low surface tension film skin of the bursting bubble (Keene et al., 2007; Woolf et al., 1987). The primary marine aerosol therefore consists of an external mixture of salt-rich and organic-rich particles (Hawkins and Russell, 2010; Wang et al., 2017).


A long-standing question about marine aerosols is the degree to which the composition of primary sea spray is affected by biological activity in the surface ocean. Recent results based on artificial bubbling of seawater in the open ocean indicate that biological productivity has a minor influence on sea spray OC and its CCN properties, suggesting that the organics associated with sea spray are derived from long-lived, well-distributed marine DOM (Bates et al., 2020; Quinn et al., 2014; Russell et al.,

2010). Beaupré et al. (2019) reported that highly aged DOM carbon could account for 19-40% of the organic carbon in artificially generated sea spray. Ceburnis et al. (2016) found that most organic enrichment in marine aerosol over the southern Indian Ocean was attributable to fresh POM. Enrichment of more bio-labile carbon compounds has also been reported in artificial seawater bubbling experiments and in ambient aerosol (Facchini et al., 2008). Artificial sea spray generated in laboratory mesocosms exhibit differences in composition related to the abundance and composition of algal and bacterial

communities (Ault et al., 2013). There is conflicting evidence regarding the influence of biological activity on the cloud condensation nucleus (CCN) activity of the generated particles (Bates et al., 2020; Collins et al., 2013; Fuentes et al., 2011; Prather et al., 2013).

There is growing evidence that biogenic polysaccharides are a component of marine aerosol. Marine phytoplankton produce

polysaccharides that become part of the seawater organic carbon pool (Orellana and Leck, 2015; Passow, 2002a). These polysaccharides may form gel-like aggregates in seawater and contribute to the transparent extracellular polymer particles (TEP) that can represent the majority of the particulate organic matter in biologically active regions of the surface ocean (Mari and Burd, 1998; Passow, 2002b). Leck and Bigg (2005) noted the presence of gel-like structures in transmission electron micrographs of dried and rehumidified marine aerosol particles across many ocean basins and proposed that these particles

were derived from marine polysaccharides. Surface ocean gels in the marine Arctic have been shown to be structurally similar



to particles found in overlying marine aerosol samples (Leck et al., 2013; Orellana et al., 2011). The presence of polysaccharides in marine aerosols is consistent with FTIR observations of a "carbohydrate-like" major component in primary marine aerosol (Russell et al., 2010). Size-resolved polysaccharides have been quantified in a study in the high Arctic, using high-performance liquid chromatography of saccharide monomers following hydrolysis of the polymers and oligomers (Leck et al., 2013). Aller et al. (2017) recently detected polysaccharide (Alcian blue stainable) material in artificially generated sea spray and one ambient sample from the western North Atlantic. They observed increasing polysaccharide fractions toward smaller particle sizes for submicron artificial primary aerosol, with contributions at the different cut points and stations ranging from < 1% to greater than 50%.

In the present work, we describe measurements of marine aerosol made applying TDCIMS (thermal desorption chemical ionization mass spectrometry) and FTIR techniques to the same samples. These techniques have the potential to detect both refractory and labile components of the organic matter in marine aerosols, and therefore contribute to our understanding of the composition and origin of this material. The samples analyzed here were collected during the North Atlantic Aerosols and Marine Ecosystems Study (NAAMES) (Behrenfeld et al., 2019). One of the main study goals was the characterization of the seasonal influence of surface ocean ecosystems on atmospheric aerosols. Four month-long scientific cruise deployments of the R/V *Atlantis* were carried out during NAAMES, in November 2015, May 2016, September 2017, and March-April 2018.

## 2 Methods

### 2.1 Ambient aerosol samples

Ambient air was sampled by an inlet at ~ 900 liters per minute (LPM) from a height of 18 m above the sea surface, based on the design described in Bates et al. (2002). The inlet was installed on top of the instrument van, which was located on the second deck, forward of the bridge. The top of the inlet was 5 m above the top of the van. Ambient aerosol was collected onto the center 10 mm of 37 mm diameter PTFE filters at three size cutoffs. Each size-resolved sample filter had a dedicated subsampling line drawing from the plenum at the base of the inlet. Size cutoffs were made using single-stage Berner impactors to isolate particles at <180 nm and < 500 nm, and a cyclone was used to isolate submicron particles. Only results from <180 nm and submicron (PM1) samples are presented here A filter downstream of each size-discriminating stage collected particles as the sample air passed through it. A typical sampling time was 23 hr for ambient samples. After sampling, filters were stored below 0°C until analysis. For a subset of samples, an additional filter was installed behind the sample filter, contacting it. These "field blanks" were intended as a measure of breakthrough and contamination from sample handling.

Ambient aerosol samples were characterized as marine (as opposed to continental or mixed) on the basis of four criteria: CN concentrations <1500 cm$^{-3}$, 48 hours back trajectories over ocean, BC <50 ng/ m$^3$, and radon <500 mBq/m$^3$ (described in more detail in Lewis et al., in prep.). Samples with abaft winds were excluded from the analysis.



## 2.2 Sea Sweep aerosol samples

Artificially generated sea spray particles were produced using Sea Sweep, a raft deployed next to the ship which generated bubble-bursting sea spray aerosol by injecting air bubbles 0.75 m below the sea surface (Bates et al., 2012). Details specific to Sea Sweep operation during NAAMES can be found in Bates et al. (2020). The aerosol was sampled into an inlet and subsampled into three independent channels. Two channels had a single stage Berner impactor (<180 nm, <1100 nm) and one channel had a cyclone to isolate submicron aerosol. Aerosol was collected on the center 10 mm of a 37 mm diameter PTFE filter downstream of each impactor. Samples were typically collected for only 2 hr due to the high aerosol loadings. Field blanks were collected for a subset of samples.

## 2.3 Transmission Fourier transform infrared spectroscopy (FTIR)

All samples were analyzed by transmission FTIR using an automated fitting algorithm and techniques described previously (Maria et al., 2002; Russell et al., 2009; Takahama et al., 2013).

## 2.4 TDCIMS extraction and analysis

A subset of the ambient aerosol and sea sweep samples analyzed by FTIR were selected for mass spectroscopic analysis by TDCIMS. When possible, all size cuts in a given sample time period were analyzed by TDCIMS. For ambient samples, periods characterized as "clean marine" were preferentially chosen. Two periods with a strong continental influence were included for comparison. The filters were extracted into 6 mL ultrapure water (Milli-Q Advantage A10, 18.2 Mohm) by sonication. Each filter was inserted into a 10 mL polypropylene centrifuge tube (Falcon), and 6 mL ultrapure water was added to cover the filter. Several tubes at a time were supported by a plastic rack in a bath-style sonicator (Fisher Scientific FS30) during 30 minutes of sonication. Sonication into water has been shown to be effective for extraction of polysaccharides (Cochran et al., 2017; Leck et al., 2013), the main target species for the analysis. Other water-soluble species should also have been efficiently extracted. In some cases the filters showed obvious coloration, usually brownish or yellowish, which was not fully or even mostly removed by sonication, indicating the presence of insoluble compounds.

After extraction, the samples were analyzed by offline thermal desorption chemical ionization mass spectrometry (referred to in this manuscript simply as TDCIMS). In regular use, TDCIMS is an in situ, semi-online technique that analyzes a sample of thousands to millions of particles collected from the air onto a Pt filament over 30-120 minutes (Lawler et al., 2018; Voisin et al., 2003). Prior to TDCIMS analysis, sample extracts were handled in a homemade acrylic hood with positive particle-free flow provided by a pump and HEPA filter to minimize the possibility of contamination. A 2 μL subsample of the aerosol filter extract was withdrawn from the centrifuge tube in the positive pressure hood using a microliter syringe, then carried to the instrument. A gas-tight window assembly on the instrument was opened to permit the direct application of the 2 μL droplet to



the Pt filament, then immediately re-closed. The filament was then dried in a flow of ultrahigh purity $N_2$ gas delivered by a high pressure liquid $N_2$ dewar for about 2 min. Removing the water prior to analysis avoids strong interference by the evaporating water on the chemical ionization process.

The subsequent analysis was the same as in situ TDCIMS, described here briefly. The filament was translated up into the ion source of the mass spectrometer, and the filament was resistively heated using a programmed current ramp up to a temperature of about 800 °C. The filament temperature was not directly measured, but the maximum filament current was chosen to enable the volatilization and detection of analytical standards of NaCl, which has a melting point around 800 °C. The volatilized extract components were ionized by $H_3O^+(H_2O)_n$ ions generated from trace water vapor in the ion source region and ion-molecule reactions initiated by a 0.5 mCi $^{210}$Po alpha source (1U400, NRD Static Control LLC). Full positive ion mass spectra from about 10-780 amu were collected using an API-TOF time-of-flight mass spectrometer (HTOF, Aerodyne). At least two replicates of each filter extract were run, and additional replicates were run in case of clear discrepancies. Sometimes large spurious signals were detected, requiring further analyses. We attribute these cases to contamination of the sample or filament during handling, most likely during the transfer of the 2 μL sample from the sample hood to the filament, when the syringe and the filament both got some exposure to lab air. In some cases these bad runs were overlooked at the time of analysis, resulting in only a single analysis for a given sample.

The TDCIMS response to polysaccharides was assessed using suspensions of known mass concentrations of nanocellulose fibers (University of Maine, Forest Bioproducts Research Institute) in ultrapure water (milliPore, 18.2 MΩ). For comparison, sodium alginate (Sigma Aldrich) was also tested. It showed very similar mass spectral response to nanocellulose, but it was harder to achieve a stable calibration curve, presumably due to its tendency to coagulate in solution. The instrument response to $Na^+$ was characterized with NaCl (Sigma Aldrich) solutions in ultrapure water. A standard of humic acid (Sigma Aldrich) was also analyzed. All ion signals were background-corrected using ultrapure water as an instrument blank.

**2.6 Data reduction by positive matrix factorization**

Positive matrix factorization (PMF) was used to isolate types of organic aerosol represented by the ions detected in the TDCIMS analysis. The software employed was EPA PMF 5.0. A subset of 99 of the roughly 1200 high resolution ions fitted to the TDCIMS data from the aerosol filter samples were selected for PMF analysis. All quality-controlled, TDCIMS-analyzed, background-subtracted samples were included in the dataset analyzed by PMF, but no analytical standards were included. Criteria were established to isolate signals with clear thermal desorption peaks and consistent signal above blanks and exclude other ions. Also, to reduce complexity, nitrogen-containing ions were excluded from this analysis. The primary target for the analysis was polysaccharides, and while these can contain nitrogen, the standard we used was not N-containing. $C_2H_5O^+$ is a notable major peak which was excluded. It was exerting too much control over the PMF analysis.



## 2.7 Cation quantification by ion chromatography

After analysis by TDCIMS and frozen storage, aerosol filter extracts of sub-180 nm and PM1 aerosol were thawed and analyzed by ion chromatography for $Na^+$, $NH_4^+$, $K^+$, $Mg^{2+}$, and $Ca^{2+}$. For each sample, a 2 mL subsample was pipetted into a 10 mL rack-compatible polyproylene vial and topped with a cross-slit cap for use with an autosampler. An ion chromatograph

(Metrohm 940 Professional IC Vario with a Metrosep C4-150/4.0 column using a 20 μL loop injection was used to analyze the samples. Ions were detected by conductivity, and MagicIC software was used to integrate peaks. The cations were calibrated between 10 and 1000 parts per billion by mass (ppb), equivalent to 0.06 to 6 μg total $Na^+$ collected mass on the filters, with standards made daily from a stock cation solution with 100 parts per million of each cation by mass (Metrohm Custom Cation Mix 2). All but one filter had $Na^+$ loadings within the range of standardization (one exception at 6.29 μg). Field

blanks showed detectable signal, at the lowest levels measured for almost all samples. No correction to the cation samples reported here was made on the basis of the field blanks.

## 3 Results

### 3.1 TDCIMS mass spectra

A wide variety of ions were detected in the TDCIMS mass spectra of the filter extracts, including sea salt cations, oxygenated

organics, and reduced nitrogen compounds (Figure 1). Based on the timing of ion evolution in the desorption thermograms, most of the detected peaks arose from thermal decomposition of the collected aerosol phase species (Figure S1.). The desorption thermograms provide some information about the temperature at which the various species were volatilized.

### 3.2 PMF analysis results

Several PMF models were generated to find factors that would best represent the data, and we settled on a model with five factors. Adding more factors led only to modest improvements in the explanatory power of the model (Figure 2). We termed the factors polysaccharide, recalcitrant, fatty acid, SOA1, and SOA2 (i.e. secondary organic aerosol types 1 and 2). The mass spectra have low similarity to one another and overall the factors have low temporal correlation. One exception is a sample with high organic loading that shows the highest SOA1 and SOA2 values, driving a correlation between those factors that otherwise is not apparent. A 4-factor analysis combined the two SOA factors into one. A 3-factor analysis combined the

polysaccharide and recalcitrant factors into one. Introducing a new factor (for 6 total) resulted in the addition of a hydrocarbon-like factor from peaks that had appeared primarily in the fatty acid factor. This appeared to be a real independent factor with a unique mass spectrum, but adding it resulted in little increased explanatory power of the model and it showed no significant relationships to the other variables of interest. A 7-factor solution gave the first clear evidence of a factor that seemed to be a



combination of two independent factors (SOA1 and polysaccharide). Based on our exploratory testing, we find that solutions
        of 4 to 6 factors would be defensible, and 5 captures the most variability using the fewest factors.

### 3.2.1 Polysaccharide factor

A major goal of the PMF analysis was to extract a quantitative polysaccharide tracer. In every factorization attempt with at
least 4 factors, at least one polysaccharide-like factor emerged. With 4-6 factors, the polysaccharide-like peaks were confined
to a single factor which matched the mass spectral features of the nanocellulose polysaccharide standard very closely, as seen
        in the chosen 5-factor solution (Figure 3). It should be pointed out that the standards were not included in the data used to
        generate the PMF factors, so this mass spectral signature arises naturally from the ambient and Sea Sweep aerosol samples.
        The PMF factor was scaled to match the largest peak ($C_5H_5O_2^+$) and the most characteristic polysaccharide peak ($C_6H_5O_3^+$).
        Treating $C_6H_5O_3^+$ as a calibration species for polysaccharides gave very similar results to using the PMF factor. The $C_6H_5O_3^+$
response of the instrument to the polysaccharide standard is linear (Figure S2.), and a linear scaling of the PMF factor was
        used to quantify aerosol polysaccharide. This is the only PMF factor for which we attempted calibration and quantification
        using a known standard (Figure 3). A handling background of 0.15 µg for each filter was subtracted on the basis of the field
        blank concentrations.

Ambient submicron polysaccharide concentrations were calculated on the basis of the PMF polysaccharide factor scaled to a
        known nanocellulose standard (Figure 4). Concentrations ranged from below the detection limit of about 5 ng m$^{-3}$ to 93 ng m$^{-3}$, or about <30 to 520 pmol m$^{-3}$. The range reported for total hydrolyzable sugars in the high Arctic submicron aerosol was 4
        to 140 pmol m$^{-3}$ (Leck et al., 2013). This should be a fair comparison to our measurement, and these results suggest that
        polysaccharide concentrations are often higher over the North Atlantic than over the Central Arctic. This is reasonable to
expect, considering the extremely low aerosol concentrations in the summertime Central Arctic, owing to limited sea spray
        production in the pack ice and relatively long atmospheric transport times from the open ocean. Concentrations of roughly an
        order of magnitude larger were reported in the western North Atlantic in May 2014, roughly 1000 ng m$^{-3}$ for submicron
        particles, with over 500 ng m$^{-3}$ below 180 nm (Aller et al., 2017).

### 3.2.2 Recalcitrant factor

The recalcitrant factor (Figure 5) was so called because it included organic peaks appearing at high temperatures, after the
polysaccharide standard peaks and prior to the NaCl peaks, indicating high resistance to thermal decomposition. We suggest
that it may be linked to the chemically recalcitrant, long-lived dissolved organic matter (DOM) present throughout the world
oceans (Hansell, 2013). Previous work using analytical pyrolysis coupled to chemical ionization mass spectrometry (direct
temperature-resolved mass spectrometry) showed that long-lived, chemically stable marine organics are likely to be thermally
        stable as well (Boon et al., 1998). The humic acid standard also contained similarly thermally stable organic compounds, but



the mass spectrum was very different, dominated by a hydrocarbon CH2-addition series starting with $C_4H_4$ up to $C_{12}H_{20}$ (Figure S3.). We therefore make no attempt to quantify the recalcitrant factor on the basis of this standard.

### 3.2.3 Fatty acid factor

This factor was identified by strong contributions from palmitic acid ($C_{16}H_{33}O_2^+$) and myristic acid ($C_{14}H_{29}O_2^+$), as well as aliphatic fragments (Figure 6). Notably, the field blanks for this factor were comparable in magnitude to the samples. This likely indicates revolatilization and subsequent deposition on the following filter. This effect has previously been observed for aliphatic alkanes with similar volatilities where about 66% of $C_{29}$ saturated alkane (melting point 63.7 °C, similar to 62.9 °C for palmitic acid) was lost from a Teflon aerosol sampling filter due to volatilization (Kavouras et al., 1999).

### 230 3.2.4 Secondary organic aerosol factors 1 (SOA1) and 2 (SOA2)

Two PMF factors were identified that appear to be components of secondary aerosols because of 1) higher levels in aerosol samples impacted by continental air, 2) a strong relationship with FTIR acid group mass, and 3) no correlation with $Na^+$ (described in more detail below). The SOA1 factor showed a homologous series of oxygenated peaks $C_6H_7O_2^+ – C_9H_{13}O_2^+$, as well as a dominant $C_5H_7O^+$ ion and some non-oxygenated fragments (Figure 7). SOA2 was strongly associated with air masses

with strong continental influence. It shows similar homologous series to the SOA1 factor, but with different major ions and a higher level of oxidation.

### 3.4 Organic factor correlations with FTIR and sodium

The five TDCIMS-derived PMF factors and TDCIMS- and IC-derived measurement of $Na^+$ were compared against the five functional groups analyzed by FTIR for the same submicron (PM1) samples. This analysis is not presented for sub-180 nm

samples because the collected mass was low and FTIR groups were frequently below detection limits. The results of the correlation analysis are reported in Tables 1 and 2. Relationships were considered strong if the slope of the regression line was at least four standard errors greater than zero, and modest if the slope was at least two standard errors greater than zero. In some cases a single sample with high signal for one or more components exerted a strong influence on the correlation analysis. In those cases a coefficient of determination was reported for the complete data set and the data set with that sample removed.

The relationship was considered strong if both the complete and reduced data sets showed a strong relationship, and modest if at least one of the slopes showed at least modest correlation.

### 3.4.1 Ambient marine PM1 organic correlations

There are some clear correlations between the TDCIMS- and FTIR-based organic analyses. The most striking association is between the TDCIMS recalcitrant factor and the FTIR alcohol and amine groups for ambient marine aerosol. All three of these

organic signals correlate strongly with $Na^+$. It is likely therefore that all of these signals originate from sea spray organics. This





is consistent with prior studies that connected FTIR alcohol and amine signals to primary sea spray (Bates et al., 2012; Russell et al., 2010).

The FTIR acid group showed no relationship to sodium. This is consistent with previous work indicating an association of this component with secondary or continental aerosol (Hawkins et al., 2010; Frossard et al., 2014). Because the acid group correlated strongly with two factors which were not associated with sodium, we inferred that the factors likely represented secondary organic aerosol (SOA). The SOA factors also showed modest relationships with the alkane group, though there was one strong outlier that influenced the correlation. In the case of SOA2, this appears to be a real relationship in that the $r^2$ was reasonably large also when the outlier was removed.

### 260  3.4.2 Sea Sweep PM1 organic correlations

The same correlation analysis between the FTIR- and TDCIMS-based organic signals and $Na^+$ was conducted for the submicron Sea Sweep samples. Similar to the ambient marine samples, $Na^+$ correlated strongly with TDCIMS recalcitrant, FTIR alcohol group, and FTIR amine group. However, the correlation coefficients were not as large, and overall for the Sea Sweep samples there was less of a clear distinction between different organic types in terms of their relationship with $Na^+$. In
the Sea Sweep samples, most TDCIMS factors and FTIR groups correlated with $Na^+$, even the SOA1 and SOA2 factors attributed to secondary aerosol. This could arise as a result of entrainment of some ambient air during Sea Sweep sampling. Any contaminant that increases with increased sample time would be expected to correlate with $Na^+$. The fractional contributions of the PMF factors suggest a small magnitude of any such effect (Figure S4.). Two components did not correlate with $Na^+$: 1) the carbonyl group, which was detected in only one of the analyzed Sea Sweep samples and 2) the fatty acid PMF
factor, which showed indications of volatilization as described above. We find that this correlation analysis supports the idea that the TDCIMS recalcitrant factor comes from primary sea spray, along with FTIR alcohol and amine groups, but beyond that we cannot infer much from the many modest relationships observed among organic signals in the Sea Sweep aerosol.

### 3.4.3 Polysaccharide correlations with inorganic cations

To investigate whether marine aerosol polysaccharides may be associated with inorganic cations, we assessed the correlations between polysaccharide and all the ion concentrations determined, for ambient and Sea Sweep aerosol, at both size cuts. Similar to the results for the comparisons to PM1 $Na^+$ in Tables 1 and 2, polysaccharide shows at least a modest positive correlation with most of the sea spray-derived cations for the Sea Sweep samples at both sizes, excepting $Ca^{2+}$ (Table 3). There is a strong correlation between polysaccharide and potassium for sub-180 nm Sea Sweep aerosol, suggesting that potassium
may be physically associated more strongly than other cations with the polysaccharides in the surface ocean which become aerosolized into fine particles. While there remains no positive relationship between polysaccharide and any cation in ambient



marine submicron aerosol, polysaccharide also has a modest positive relationship with $K^+$ in ambient marine sub-180 nm aerosol.

### 3.5 Comparison with previous TDCIMS-based marine ultrafine aerosol observations

The in situ TDCIMS technique has been applied to marine aerosol in one previous published study (Lawler et al., 2014).That study focused on the composition of sub-100 nm aerosol measured at the coastal site Mace Head in Ireland, and particularly on events during which Aitken mode particles increased substantially in number. Similar to the present study, the acetaldehyde ion was the major organic ion detected in positive mode and also attributed to the decomposition of larger species. It was not connected to the Aitken mode events, but the ions $C_7H_7O_2^+$ and $C_9H_{19}O_2^+$ were. These were attributed to benzoic acid and nonanoic acid, respectively. The Aitken mode events were not linked to either sodium or sulfate, making it difficult to determine whether the particles were primary or secondary. It was concluded that new particle formation was the most likely explanation primarily on the basis that sea spray source functions should result in a broader distribution of observed aerosol sizes. No PMF analysis was attempted for that study due to the limited number of detectable peaks.

We investigated the distribution of $C_7H_7O_2^+$ and $C_9H_{19}O_2^+$ in the NAAMES samples by examining the correlation to $Na^+$ and the TDCIMS PMF factors (Table 4). On the basis of these results, $C_2H_5O^+$ appears to be a nonspecific ion that arises from the ambient aerosol but cannot readily be linked to one of the identified aerosol types. Both $C_7H_7O_2^+$ and $C_9H_{19}O_2^+$ show some association with the recalcitrant factor and $Na^+$, and so are likely present in primary aerosols. $C_7H_7O_2^+$ appears to have more than one source, as it also strongly correlated with SOA2 and modestly correlated with SOA1. $C_9H_{19}O_2^+$ was not connected to either of the SOA factors, implying that it was not formed in the same secondary processes. The formula $C_9H_{19}O_2^+$ is consistent with a fatty acid, so it is understandable that it correlates at least modestly with the fatty acid factor. On the basis of these results it is difficult to substantially revise the conclusions of the Mace Head study.

### 3.6 Seasonal relationships among organic and inorganic aerosol components

The polysaccharide PMF factor did not correlate strongly with any aerosol component in either the ambient or Sea Sweep PM1 aerosol. There was a modest correlation between polysaccharide and $Na^+$ in Sea Sweep aerosol, but that was true of most of the PMF organic factors, even the secondary aerosol factors. To explore the relationship of aerosol polysaccharide to primary sea salt and to attempt to identify seasonal patterns in the relative fraction of polysaccharide in marine aerosol, we plotted the distribution of the ratio of polysaccharide to $Na^+$, organized by season for both sub-180 nm and PM1 aerosol (Figure 8a.,e.). The PM1 data show no apparent seasonal trend in either Sea Sweep or ambient data. The ambient ratios showed a wider range of values, shifted toward higher ratios. The sub-180 nm samples also tended to have higher polysaccharide:$Na^+$ ratios in ambient compared to Sea Sweep aerosol, and this was particularly clear in the late spring and summer, indicating that these smaller particles may have a seasonal cycle in polysaccharide:inorganic sea spray ratio.



315

The seasonal patterns in potassium:sodium ratio (Figure 8b.,f.) match those of the polysaccharide:sodium ratio remarkably well, for both ambient and Sea Sweep aerosol. This is in contrast to magnesium:sodium, which is basically invariant over the year in both aerosol size classes and shows little to no difference between ambient and Sea Sweep aerosol (Figure S5.). In fact, the correlation between magnesium and sodium over all sample types is extremely good (Table S1.). Calcium:sodium was highly variable, and there was a large enrichment in calcium above seawater ratios, especially for the sub-180 nm aerosol (Table S1., Figure S6.), similar to what has been observed in previous sea spray generation experiments (Salter et al., 2016).

The PMF recalcitrant factor was closely correlated to sodium and FTIR alcohol and amine groups, particularly in ambient air, and we have hypothesized that it likely represents long-lived marine organics in primary aerosol. For PM1 aerosol, the recalcitrant factor showed a very consistent relative signal intensity to sodium mass for Sea Sweep and ambient aerosol, across all seasons (Figure 8g.). This is consistent with an organic component that is primary and ubiquitous. The sub-180 nm aerosol showed some enhancement of recalcitrant factor with respect to sodium during September, particularly for ambient aerosol. However, even in this period, there was significant overlap with respect to the primary Sea Sweep aerosol. We think it is reasonable to conclude that either the generation or atmospheric processing of the detected polysaccharides is different from the sodium-linked recalcitrant factor, based on the stronger and more consistent ambient-Sea Sweep differences observed for the polysaccharide ratio.

To understand the relative fraction of total organics the polysaccharides represent, the mass ratio of polysaccharide to total FTIR organics was plotted across all cruises for PM1 ambient and Sea Sweep aerosol (Figure 9a.). The median or mean values ranged from about 10-30%. Fractions above unity resulted from samples for which the alcohol group was not detected by FTIR. Usually this was the major functional group detected. Since the polysaccharides are polyols, they should be detected as alcohol groups. For points with detectable polysaccharide and alcohol, polysaccharides represented typically around 20% of the alcohol group mass, with three samples showing values around 50% or higher, all in September (Figure 9b.). Because most samples of sub-180 nm aerosol were below detection limits for many FTIR organics, we do not report an analogous analysis for those smaller particles.

Total sampled primary aerosol mass was estimated by summing the FTIR-derived alcohol and amine group mass and the IC-derived cation masses. A seawater-equivalent mass of Cl⁻ was added to estimate the mass contribution of anions. This primary aerosol mass estimate was used to calculate the mass fraction of polysaccharide in primary ambient and Sea Sweep aerosol across the cruises (Figure 8d.,h.). The seasonal pattern is very similar to the polysaccharide : Na⁺ ratio, perhaps not surprising since all the primary components correlate closely with Na⁺. Higher fractions, up to about 32%, were measured in late spring



and summer in sub-180 nm aerosol for both Sea Sweep and ambient aerosol, though there was a wide range down to very low fractions of a couple percent in Sea Sweep aerosol.

These estimates can be directly compared to the TEP: total aerosol mass ratio reported in Aller et al. (2017), which focused primarily on North Atlantic Sea Sweep-generated aerosol from May of 2014. In that study polysaccharides contributed from 4-20% to 100-180 nm particle mass, similar to the sub-180 nm Sea Sweep observations from the present study. For 560-1000 nm particles they reported polysaccharide mass fractions ranging from 0.7-30%, with many values considerably larger than the PM1 Sea Sweep results from the present study. Aller et al. (2017) also reported results for one ambient aerosol sample

from May 2014 that contained roughly ten-fold higher polysaccharide concentrations than found in this study. For that sample the polysaccharide mass fraction in 560-1000 nm particles was about 13%, roughly similar to the polysaccharide: estimated primary mass ratio for submicron aerosol in this study. They reported much higher ambient fractions at 100-180 nm (~70%) than observed in any samples from this study. The discrepancy would be even larger if we included secondary aerosol mass in our estimates.

**4 Discussion**

**4.1 The components of primary marine organic aerosol**

This study used a novel approach to identify two mass spectral signatures of primary marine organics and explored their relationships to organic functional groups and inorganic cations. A key finding is that the organic material that closely resembled polysaccharide standards was not correlated with the FTIR alcohol (or hydroxyl) functional group. As in previous

studies, the FTIR results from this study indicate that the alcohol functional group is a major contributor to primary sea spray organic mass. While polysaccharides should represent some fraction of the alcohol group mass, they do not appear to be the main contributor. Instead, the TDCIMS recalcitrant factor was closely correlated to the alcohol group mass. This factor was composed of organic material that thermally decomposed at temperatures higher than polysaccharides. This thermal stability is suggestive that the recalcitrant factor (and by extension the correlated FTIR organics) could represent long-lived, refractory

marine dissolved organic material, but further work is needed to establish this connection. This will require the identification of appropriate analytical standards for the TDCIMS recalcitrant factor.

These results shed some light on the question of the importance of "fresh" compared to "aged" marine organic matter in forming the marine aerosol. Polysaccharides are thought to be a relatively dynamic component of marine organic matter in seawater, undergoing conversion from colloids to larger aggregates on timescale of days or weeks (Wurl et al., 2011). This

contrasts with the recalcitrant dissolved organic matter which may persist in the ocean for millenia (Hansell, 2013). We have found that polysaccharides represent around 10-30% of the submicron organic aerosol mass (and about 3-12% of total submicron aerosol mass) in both ambient marine and artificially generated sea spray aerosol. This fraction appears relatively invariant across seasons. However, for sub-180 nm particles, which represent the most important population of cloud





condensation nuclei (CCN) because of their higher number concentrations, there is evidence for higher polysaccharide:Na$^+$
and polysaccharide:total estimated mass ratios in the spring and summer compared to late fall (November), suggestive of a
seasonal biological influence on aerosol at these sizes. Whether this variability could impact CCN concentrations is likely to
be strongly dependent on the mixing state of the primary sea spray aerosol organics. The strongest identified determinant of
CCN concentration over the NAAMES mission was the presence of sulfate in sub-180 nm particles (Quinn et al., 2019). Sea
spray aerosol was only found to correlate with CCN in the winter during NAAMES, but sea spray was only considered to be
an internally mixed NaCl aerosol. About half of the impact of sulfate arose from its condensation onto pre-existing particles
(Sanchez et al., 2018), which could include externally mixed primary polysaccharides.

## 4.2 The origin of polysaccharides in North Atlantic aerosol

A surprising result from this study is that the relationship between polysaccharide and sea salt is very different in ambient
aerosols compared to Sea Sweep samples. The ambient aerosol shows high polysaccharide:Na$^+$ ratios and no discernible
correlation between the polysaccharides and Na$^+$, while Sea Sweep exhibits low ratios and modest correlations. These
differences are unexpected given that the polysaccharides are assumed to be derived from sea spray. These puzzling differences
are not observed for the recalcitrant organics for which the ambient and Sea Sweep samples are similar in terms of both their
ratio to and correlation with Na$^+$. The differences in behavior between the polysaccharides and the recalcitrant organics suggest
that either their distributions or their aerosol generation mechanisms (or both) are very different. This is perhaps not surprising,
given that the recalcitrant organic material in seawater is dissolved and well-mixed, while the polysaccharides are likely
particulate and/or colloidal and heterogeneously distributed due to their bioavailability and tendency to coagulate.

If polysaccharides and sea salt components of the marine aerosol are externally mixed, an explanation could be sought
involving differences in atmospheric loss rates. For example, polysaccharides may be predominantly concentrated in film
droplets, while the salts are predominantly in jet droplets. There is some evidence for this in that stronger correlations were
found between polysaccharides and inorganic ions for sub-180 nm than for submicron aerosol, in particular for K+. Previous
work has shown that gel-derived aerosol particles contain different cations depending on their morphology, with smaller
particles of 10s of nm diameter or their aggregates containing more Na$^+$ and K$^+$ than Mg$^{2+}$ and Ca$^{2+}$ (Hamacher-Barth et al.,
2016). Larger marine aerosol particles of 0.5 – 8.4 µm diameter have also been shown to have polysaccharide-like coatings
that contain K$^+$, the only inorganic cation detectable in that study (Hawkins and Russell, 2010). In a mesocosm wave flume
experiment, enrichments in aerosol K$^+$, Mg$^{2+}$, and Ca$^{2+}$ were observed in association with aerosol enrichments in oligo- and
polysaccharides for sub- 2.5 µM and sub-10 µM sea spray particles (Jayarathne et al., 2016). Ambient polysaccharide
concentrations in the present study were comparable for sub-180 nm and submicron aerosol, despite much higher total
inorganic and organic mass in submicron aerosol, suggesting that most of the polysaccharide mass was present in sub-180 nm
particles. This is generally consistent with observations of size resolved polysaccharides in the subpolar and high Arctic (Leck
et al., 2013) and in one aerosol sample from the western North Atlantic (Aller et al., 2017). Presumably, the larger, saltier jet



drop particles are more efficient CCN and more likely to form cloud droplets that are readily removed from the atmosphere as precipitation, while the less hygroscopic polysaccharide-enriched particles could accumulate. This leaves open the possibility that such polysaccharide-enriched particles could serve as seeds onto which sulfate can condense to form CCN, but determining

whether this is possible requires a better understanding of the life cycle and mixing state of polysaccharide aerosol.

**Data availability**

All NAAMES mission data are archived here by year: https://www-air.larc.nasa.gov/missions/naames/index.html, and all presented data are from the R/V Atlantis Ship platform. For specific requests, please contact Mike Lawler (mlawler@uci.edu).

**Acknowledgments**

The authors thank the crew and science techs of the RV Atlantis, Prof. Jim Smith for use of the TDCIMS, and Irmak Sengur and Michelia Dam for running test samples. This work was supported by NASA grant NNX15AF31G. This is PMEL contribution number 5110. LMR acknowledges NASA grant NNX15AE66G.

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

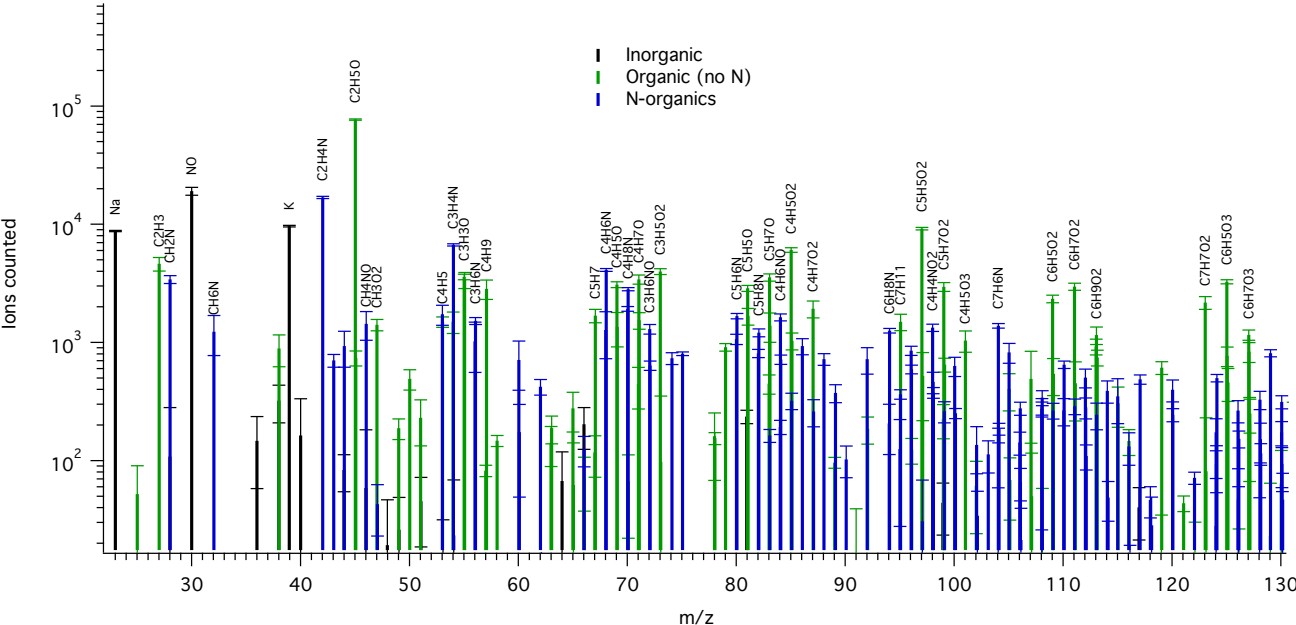

**Figure 1: Example mass spectrum of background-corrected TDCIMS-derived positive ion signals for an ambient submicron aerosol sample collected UTC 15:15 May 26 – 09:00 May 27, 2016. Ion types are color-coded: inorganic (black), organic with no N (green), and N-containing organics (blue). All labeled organic ion compositions can be assumed to include an H$^+$ adduct. Only a subset of the mass range is shown for clarity. Peaks of over 2000 ion counts are labeled, except when overlapping with a higher peak. The square root of counted ions is plotted as error bars.**

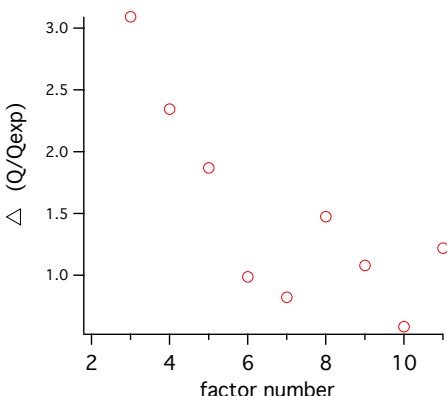

**Figure 2: Change in the ratio of Q to Q$_{expected}$ as a result of increasing to a given factor number. Relatively strong improvements are seen for changes up to five factors, with additions to 6 and 7 factors showing much smaller improvement. Above 5 factors the ratio is variable and improvement is marginal.**





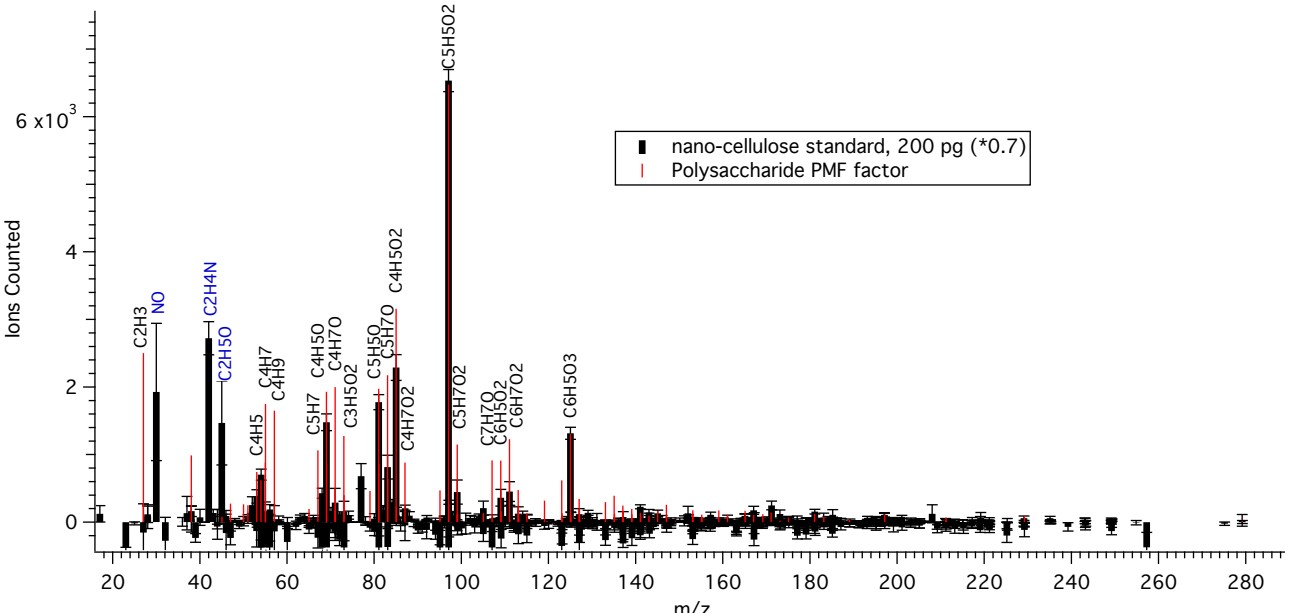

**Figure 3: Mass spectrum of nano-cellulose standard (black sticks) and the polysaccharide PMF-derived factor (red sticks) attributed to polysaccharidal aerosol material. The peaks $C_6H_5O_3^+$ and $C_5H_5O_2^+$ were used to scale the PMF factor to the standard for calibration. $C_6H_5O_3^+$ was the most characteristic polysaccharide peak and $C_5H_5O_2^+$ was the major polysaccharide peak. This 2 µL 0.1 mg/L standard is equivalent to 0.6 µg collected on a filter and to an ambient sample with an air polysaccharide concentration of roughly 0.05 µg m$^{-3}$. Large peaks in the standard which were excluded from the PMF analysis are labeled in blue (N-containing species and over-dominant $C_2H_5O^+$).**





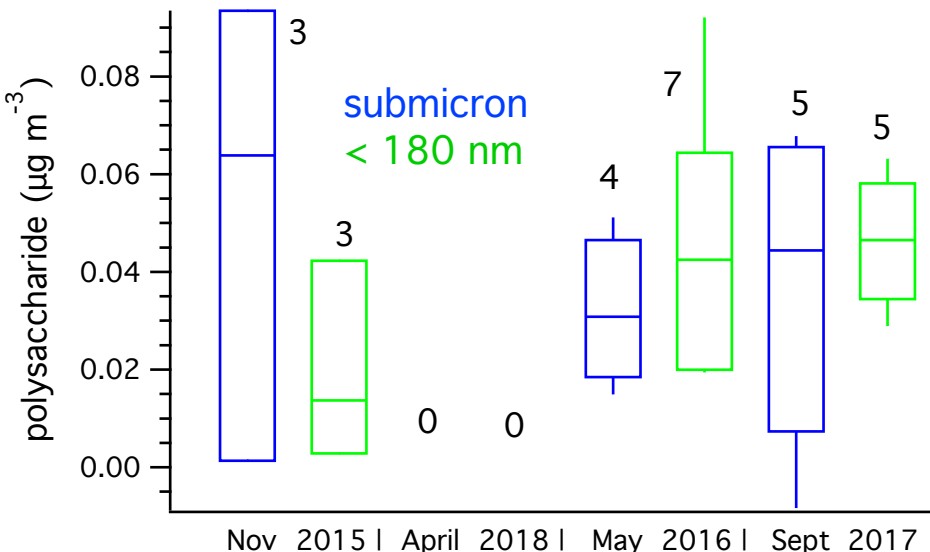


**Figure** 4**: Ambient concentrations of polysaccharide in submicron (blue) and sub-180 nm (green) aerosol determined in this study over the 4 NAAMES cruises, organized by season, with median, 25th and 75th percentiles as horizontal lines in boxes, and 10th and 90th percentiles indicated by vertical lines outside boxes. The number of samples contributing to each sample type is given above the box. Any negative (below background) signals were removed from the analysis. The similar absolute values in the two size fractions imply that most of the polysaccharide was present in sub-180 nm particles.**


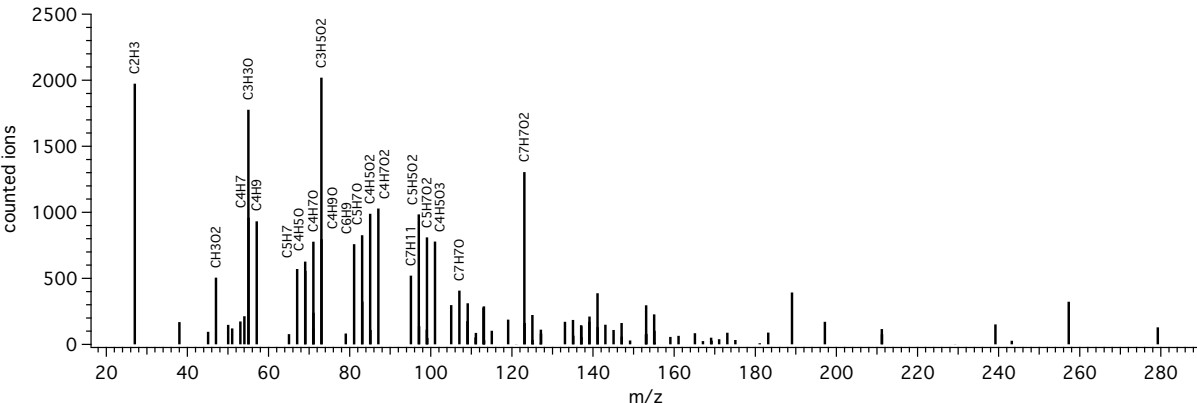

**Figure 5: Mass spectrum of the recalcitrant factor derived from the PMF analysis of TDCIMS ion signals of ambient and Sea Sweep aerosol. The most characteristic feature of this factor is a $C_7H_7O_2^+$ peak consistent with benzoic acid.**



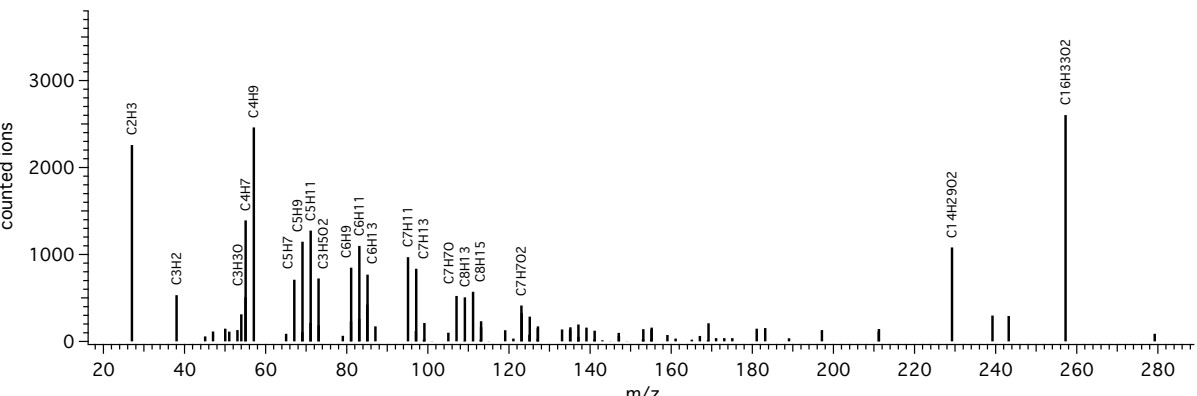

**Figure 6: Mass spectrum of the fatty acid factor derived from the PMF analysis of TDCIMS ion signals of ambient and Sea Sweep aerosol. This factor is characterized by strong $C_{14}H_{29}O_2^+$ and $C_{16}H_{33}O_2^+$ peaks and several hydrocarbon fragments.**

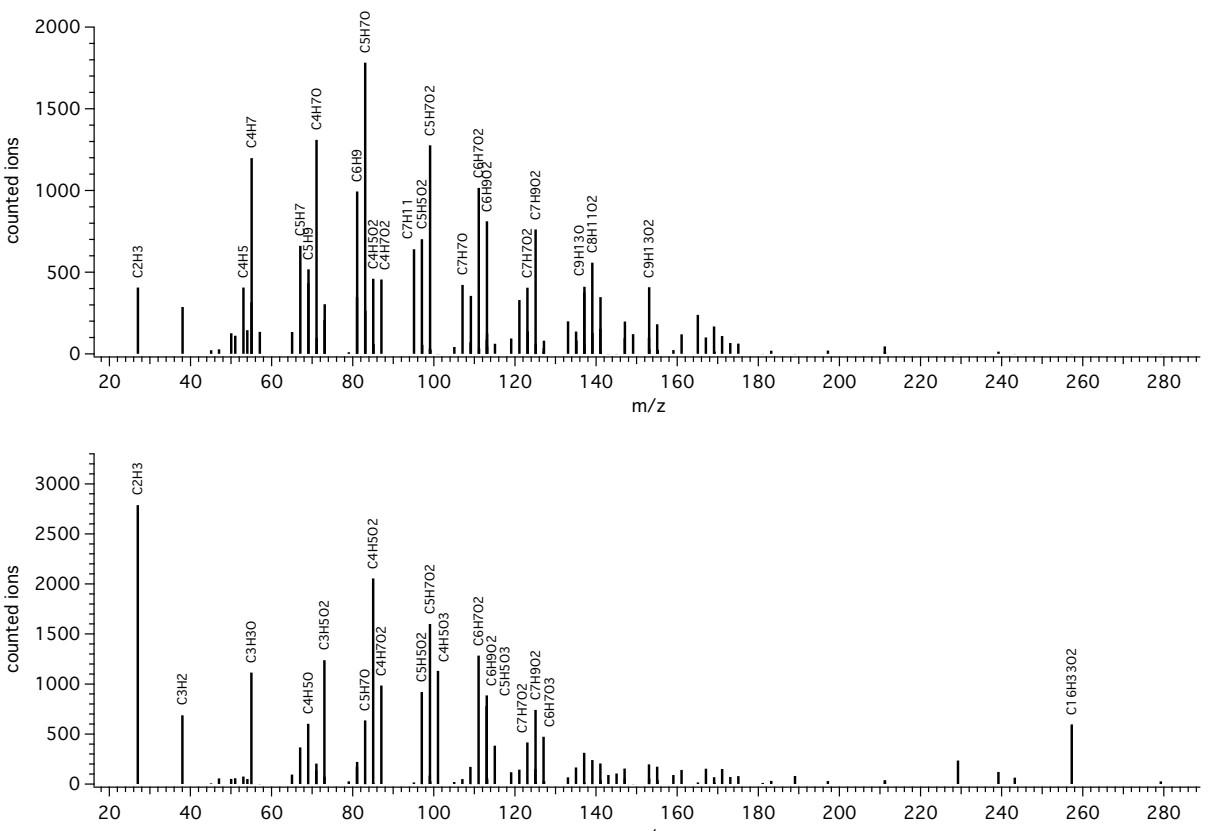

**Figure 7: Mass spectra of the SOA1 (upper) and SOA2 (lower) factors derived from the PMF analysis of TDCIMS ion signals of ambient and Sea Sweep aerosol. These factors are attributed to secondary aerosols.**



**Table 1: Coefficients of determination (r²s) for relationships between TDCIMS PMF factors, Na⁺, and FTIR organic functional groups for ambient submicron aerosol characterized as marine. Dark green squares indicate a strong statistical relationship, with a linear slope 4 standard deviations greater than 0. Light green indicates a likely relationship, with a linear slope 2 standard deviations greater than 0. Cases for which removing a single point changes a relationship between none and modest or none and strong are shown as "with (without)" that point and are colored light green. Relationships for which r is negative are indicated with (-). One anticorrelated pair was a linear slope 2 standard deviations less than zero is indicated with a light red color.**

| | | FTIR functional groups | | | | | Ion (IC) |
|---|---|---|---|---|---|---|---|
| | | acid | alcohol | carbonyl | amine | alkane | Na⁺ |
| TDCIMS PMF Factors | recalcitrant | 0.04 | 0.86 | 0 | 0.79 | 0.03 | 0.81 |
| | SOA 1 | 0.87 | (-)0.00 | 0 | (-)0.05 | 0.85 (0.07) | (-)0.02 |
| | polysacch | 0.09 | (-)0.03 | 0 | (-)0.11 | 0.10 | (-)0.04 |
| | fatty acid | (-)0.36 | (-)0.00 | 0 | 0.00 | (-)0.28 | (-)0.02 |
| | SOA 2 | 0.86 | 0.07 | 0 | 0.01 | 0.74 (0.26) | 0.04 |
| Ion (IC) | Na⁺ | (-)0.00 | 0.87 | 0 | 0.92 | (-)0.00 | 1 |






**Table 2: Coefficients of determination ($r^2$s) for relationships between TDCIMS PMF factors, TDCIMS Na$^+$, and FTIR organic functional groups for Sea Sweep-generated submicron aerosol. Dark green squares indicate a strong statistical relationship, with a linear slope 4 standard deviations greater than 0. Light green indicates a likely relationship, with a linear slope 2 standard deviations greater than 0. Cases for which removing a single point changes a relationship between none and modest or none and strong are shown as "with (without)" that point and are colored light green. Relationships for which r is negative are indicated with (-).**

| | | FTIR functional groups | | | | | Ion (IC) |
| --- | --- | --- | --- | --- | --- | --- | --- |
| | | acid | alcohol | carbonyl | amine | alkane | Na$^+$ |
| TDCIMS PMF Factors | recalcitrant | 0 | 0.15 (0.40) | (-)0.05 | 0.25 (0.06) | 0.03 | 0.55 |
| | SOA1 | 0 | 0.10 (0.21) | (-)0.01 | 0.21 (0.03) | 0.03 (0.19) | 0.41 |
| | polysacch | 0 | 0.01 | (-)0.10 | 0.06 | 0.01 | 0.18 |
| | fatty acid | 0 | 0.01 | (-)0.04 | (-)0.01 | (-)0.08 | (-)0.00 |
| | SOA2 | 0 | 0.07 (0.22) | (-)0.01 | 0.20 (0.03) | 0.03 | 0.36 (0.14) |
| Ion (IC) | Na$^+$ | 0 | 0.57 | (-)0.09 | 0.67 | 0.18 | 1 |



**Table 3: Coefficients of determination ($r^2$s) for relationships between polysaccharide mass and inorganic cation mass, for sub-180**
**and submicron samples of both ambient marine and Sea Sweep aerosol. Dark green squares indicate a strong statistical relationship,**
**with a linear slope 4 standard deviations greater than 0. Light green indicates a likely relationship, with a linear slope 2 standard**
**deviations greater than 0. Cases for which removing a single point changes a relationship between none and modest or none and**
**strong are shown as "with (without)" that point and are colored light green, and relationships for which r is negative are indicated**
**with (-).**

| | Sample type | IC cations | | | | |
|---|---|---|---|---|---|---|
| | | Na+ | NH4+ | K+ | Mg2+ | Ca2+ |
| TDCIMS polysaccharide | <180 nm marine | 0.21 (0.34) | 0.03 | 0.20 (0.32) | 0.17 | (-)0.03 |
| | <180 nm Sea Sweep | 0.29 | 0.90 (0) | 0.57 | 0.29 | 0.08 |
| | PM1 marine | (-)0.01 | 0.02 | (-)0.04 | (-)0.05 | (-)0.03 |
| | PM1 Sea Sweep | 0.16 (0.25) | 0.00 | 0.21 (0.12) | 0.16 (0.24) | 0.17 |








**Table 4: Coefficients of determination ($r^2$s) for relationships between selected TDCIMS ions (previously linked to marine aerosol in Lawler et al. 2014), TDCIMS derived PMF factors, and two Na+ measurements for ambient submicron (PM1) aerosol. Dark green squares indicate a strong statistical relationship, with a linear slope 4 standard deviations greater than 0. Light green indicates a likely relationship, with a linear slope 2 standard deviations greater than 0. Cases for which removing a single point changes a relationship between none and modest or none and strong are shown as "with (without)" that point and are colored light green.**


|  |  | TDCIMS ions | | |
|---|---|---|---|---|
|  |  | $C_2H_5O^+$ | $C_7H_7O_2^+$ | $C_9H_{19}O_2^+$ |
| TDCIMS PMF factors | recalcitrant | 0.13 (0.36) | 0.19 (0.49) | 0.39 (0.21) |
|  | SOA1 | 0.09 (0.41) | 0.85 (0.15) | 0.04 |
|  | polysacch | (-)0.02 | 0.14 | 0.16 (0.35) |
|  | fatty acid | 0.07 | (-)0.14 | 0.13 (0.32) |
|  | SOA2 | 0.11 | 0.84 | 0.05 |
| Ion(IC) | Na$^+$ | (-)0.00 | 0.02 (0.46) | 0.27 |





**Figure 8: Box-and-whisker plots of aerosol component ratios for samples taken over the 4 NAAMES cruises, organized by season: a.,e.: polysaccharide: Na+ mass ratio, b,f: K+:Na+ mass ratio (seawater ratio of 0.036 plotted as dashed line), c,g: recalcitrant factor:Na+ (arbitrary unit ratio), d.h.: polysaccharide mass: estimated total primary mass ratio. Left column plots (a.-d.) are for sub-180 nm samples, right column plots (e.-h.) for submicron samples. Median, 25th and 75th percentiles are plotted as horizontal lines in boxes, and 10th and 90th percentiles areindicated by vertical lines outside boxes. The number of samples contributing to each sample type is given above the box. Any negative (below background) signals were removed from the analysis.**



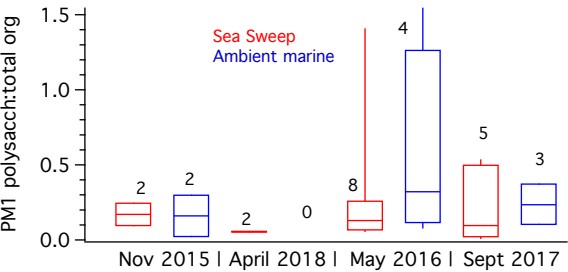 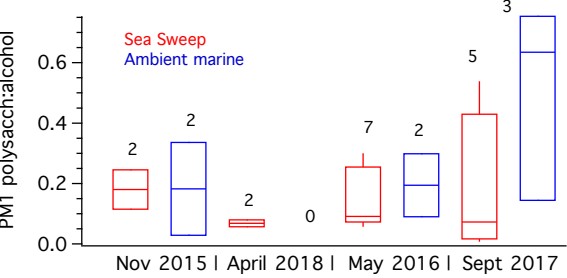

**Figure 9: Box-and-whisker mass ratio plots of (left) TDCIMS polysaccharide to FTIR total organics and (right) TDCIMS polysaccharide mass to FTIR alcohol group mass ratios for PM1 aerosol filter samples taken over the 4 NAAMES cruises, organized by season, with median, 25th and 75th percentiles as horizontal lines in boxes, and 10th and 90th percentiles indicated by vertical lines outside boxes. The number of samples contributing to each sample type is given above the box. Any negative (below background) signals were removed from the analysis.**