# Peer review of "North Atlantic marine organic aerosol characterized by novel offline thermal desorption mass spectrometry approach: polysaccharides, recalcitrant material, secondary organics"

_Atmospheric Chemistry and Physics, 2020_

## Referee Comment (RC1) · Anonymous Referee #1 · 30 Aug 2020

The study by Lawler et al. describes chemical measurements of aqueous extracts of filter samples collected during ship-board studies in the North Atlantic. Samples include both ambient and freshly-produced aerosols from the ocean surface using a Sea Sweep. The study adds to our knowledge of marine aerosol by providing further chemical analysis to support the classification of the aerosol into sub-types, including polysaccharides, fatty acids, SOA, and recalcitrant marine organic material. The discussion section of the paper focuses on polysaccharides. It is not clear why the study gives rather short schrift to fatty acids, which are an important group of marine organic

compounds that are in need of further study. The majority of the paper is devoted to the detailed description of sub-types assigned to the aerosol samples, and so is largely descriptive. The fundamentally important results of this paper should be made clearer and more explicit. The discussion section highlights at least one unexpected result, which is not described in the abstract, nor highlighted and analyzed adequately within the context of the existing literature. While the paper appears technically sound, it could stand to be revised. Sampling a wider swath of the existing field and laboratory studies of sea spray and marine aerosol would provide for a more comprehensive discussion and contextualization of the measurements. This review also questions the manner in which the four sampling periods are interpreted as representatives of seasonal variability.

This paper is likely to be publishable in Atmospheric Chemistry and Physics upon minor revision.

Figure 8: I am concerned about the presentation of this data and its interpretation. The use of box and whisker plots to represent a dataset with a small number of samples may be leading to suspicious interpretations – or at least could confuse the interpretation of the data. What does a box and whisker plot mean when only three samples are included? The box clearly no longer represents an inter-quartile range. . . because how does one have quartiles with three samples? Or in panel (a), there is a group of 4 samples with a median value close to the lower end of the interquartile range, so is the 75th percentile and 90th percentile driven by just one point with a high ratio? It may be more useful to use a violin plot (showing the distribution of data rather than a box) or simply a plot of all of the data points without statistical treatment since the number of samples is so small. At the same time, while the authors clearly have a few data points for April, presenting the lack of data may confuse the interpretation. To be clear, it is understandable that only a small number of samples exist – the question is really about data reduction and associated interpretation.

In addition, it would be helpful to the reader's interpretation of the data if some brief description of the conditions of the sampling location and/or general trends in biological productivity are described for each of the months shown. A reader who is less initiated with respect to the general seasonality or sampling plans for these North Atlantic cruises would benefit significantly from a short description of the conditions.

Lines 382 – 386: The early part of this set of lines is understandable with respect to the suggested association between CCN and sulfate. But then in the last line of the paragraph, polysaccharides are once again invoked. This transition back to polysaccharides is confusing. Is there evidence for polysaccharides to be in Na-free sea spray particles? Studies on this topic have been conducted. The question is not that this is speculative, but that the speculation is not well contextualized or supported by outside evidence.

Lines 398 - 400: See commentary about evidence for aerosol mixing state above. Sampling the literature on the mixing state of sea spray aerosol more thoroughly would improve this discussion substantially.

Minor comment Line 110: should be "mass spectrometric" as mass spectroscopy is not the proper name for mass spectrometry

---

## Referee Comment (RC2) · Anonymous Referee #2 · 3 Sep 2020

The study by Lawler et al. is dedicated to applying state-of-the-art instrumentation to elucidate sea spray composition over the North Atlantic with a particular focus on transferrable water soluble organic species and their relationship to ocean biological activity. The main idea of the study to utilise TDCIMS is innovative and producing new insights into the subject. The main problem with the paper and forthcoming conclusions is the number of samples which makes it difficult in arriving at robust inferences due to inherent natural variability temporally as well as spatially. The authors did their very best in trying to squeeze as much information as possible from the data analysis

and fairly interpreting their findings. However, due to scarcity of data the conclusions are rather vague and tend to overly rely on previous literature findings and seeking commonality, which is only natural given limited sample bank. I am not sure if the paper should rather be published in Atmospheric Measurement Techniques which would be a more suited medium for application of innovative techniques and somewhat limited dataset. ACP is instead reporting scientific advancements which are quite limited in this study due to overarching limitations. As an example of difficulties the authors confronted in terms of robustly derived conclusions is poor statistics. How can basic statistical variables be derived from e.g. 3 samples? Overall, the paper is written very well and follows clear structure, so apart from vague conclusions (not of authors fault) can possibly be published in ACP after addressing the comments.

Comments in their sequence:

Title I think the title should be amended to "...thermal desorption mass spectroscopic approach:..." Later in the methods it is worded exactly that way.

Line 89. It is unclear whether submicron fraction overlapped with <180 nm fraction or was it separate 180-1000nm size fraction?

Line 95. The criteria are appropriate, but not very conservative. Typically, CN concentrations in clean marine atmosphere are <1000 cm-3, except for new particle production events. Radon concentration is typically more like <200mBq/m3 and BC<30ng/m3 (but depends on the measurement method and instrument). While the differences are not large, but it is important to observe that the increase in CN is not associated with the increase in BC or radon, which would immediately suggest diluted continental outflow or ship own exhaust. 48 hour trajectories are good, but should be checked if they stay in the boundary layer or were affected by large scale subsidence. In summary, I would argue for a more conservative and integrated approach, because typical marine background is very low and can be easily perturbed by even by sporadic pollution events including own ship (stacks or various exhausts).

Line 110. How many samples a subset consisted of? It is not clear how many "clean marine" samples were preferentially chosen. Later in the text and figures can be seen that probably 27 "clean marine" samples were analysed in two size fractions each. How many corresponded to Sea sweep and how many to the ambient? All that information should be stated right at the beginning. At the moment the sample set is completely under described.

Line 149. NaCl fraction in sea salt is about 70%. What was the reason not choosing sea salt standard from Sigma-Aldrich or similar company? Na may respond differently in a more complex matrix in the instrument, but having sea salt matrix would certainly be compatible with sample extracts.

Line 153. PMF value and purpose was developed for large datasets/matrixes. Having only a handful of samples what was the purpose of statistical multivariate technique, when it is well known that PMF output (and of every statistical technique) is more reliable with large number of samples? I believe rich spectroscopic signatures were the reason to analyse the results statistically, but still highly non-symmetrical matrix would hardly produce reliable results.

Line 241. It is not stated that the correlations were statistically significant at whatever level which is especially important for a small dataset. Only then one can consider strong or modest relationships based on slopes or coefficient magnitude.

Line 264. It could well be a consequence of sea sweep device generating abnormal sea spray concentrations (Na) compared to wave breaking of plunging jets/sheets reminiscent of real waves. Correlating very large numbers with very small can disturb the pattern due to the variability of each.

Line 385. What does internally mixed NaCl represent? NaCl cannot be considered separately from other sea salt ions all of which simultaneously make sea salt. Sea salt can be internally mixed with organics and if organics is primary then both are termed sea spray.

Line 396. Coagulation is extremely unlikely given very low particle numbers in clean marine air. Textbook coagulation requires many orders of magnitude higher aerosol concentrations. Sea sweep is probably different in relation to number concentration, but in any event sea sweep is poor representation of real world wave breaking produced sea spray.

Line 398. All sea spray aerosol is born as a sea water droplets. Organic matter species cannot be taken out of sea water in a dry state. Not even concentrated microlayer film can be without some amount of sea water. Therefore, it is inevitable that at least a small amount of sea salt would be present in an aerosol particle even if organic species would overwhelmingly dominate aerosol particle volume. As a consequence, primary sea spray components cannot be considered externally mixed, because they are variants of internal mixture. Only secondary components can be externally mixed, because they originate via different process (gas-to-particle conversion).

Figure 4. How can meaningful statistics be extracted having 3-5 samples in each type? Even 7 samples is barely enough. Presenting average and a range is all can be done here. September submicron samples suggest negative polysaccharide concentration - what does that mean?

[Figure]

---

## Author Comment (AC1) · 14 Oct 2020

Author responses in blue following reviewer comments. Changes to the manuscript are italicized.

Referee 1 Comments

The study by Lawler et al. describes chemical measurements of aqueous extracts of filter samples collected during ship-board studies in the North Atlantic. Samples include both ambient and freshly-produced aerosols from the ocean surface using a Sea Sweep. The study adds to our knowledge of marine aerosol by providing further chemical analysis to support the classification of the aerosol into sub-types, including polysaccharides, fatty acids, SOA, and recalcitrant marine organic material. The discussion section of the paper focuses on polysaccharides. It is not clear why the study gives rather short schrift to fatty acids, which are an important group of marine organic compounds that are in need of further study. The majority of the paper is devoted to the detailed description of sub-types assigned to the aerosol samples, and so is largely descriptive. The fundamentally important results of this paper should be made clearer and more explicit. The discussion section highlights at least one unexpected result, which is not described in the abstract, nor highlighted and analyzed adequately within the context of the existing literature. While the paper appears technically sound, it could stand to be revised. Sampling a wider swath of the existing field and laboratory studies of sea spray and marine aerosol would provide for a more comprehensive discussion and contextualization of the measurements. This review also questions the manner in which the four sampling periods are interpreted as representatives of seasonal variability.

This paper is likely to be publishable in Atmospheric Chemistry and Physics upon minor revision.

We agree with the reviewer that fatty acids in the marine environment require further study. They were given little attention in this manuscript because we could not discern a clear particle phase signal above the background, as stated. We have modified that statement to clarify:

"This likely indicates a *major gas phase contribution* or revolatilization and subsequent deposition on the following filter. *Atmospheric palmitic and myristic acids can have higher gas phase than particle phase concentrations (Cautreels and Van Cauwenberghe, 1978; Kavouras and Stephanou, 2002). In addition, a revolatilization artifact* has previously been observed for aliphatic alkanes with similar volatilities where about 66% of $C_{29}$ saturated alkane (melting point 63.7 °C, similar to 62.9 °C for palmitic acid) was lost from a Teflon aerosol sampling filter due to volatilization (Kavouras et al., 1999). *Our results show that fatty acids were present, but it is not clear whether the present technique provides a reliable proxy for their concentration in aerosol. The results for this factor are included for completeness."*

We find the most important results to be the quantification of marine aerosol polysaccharides, the revelation that the FTIR alcohol group identified in marine aerosol is not primarily polysaccharide, and the apparent size- and season-dependence of polysaccharide aerosol mass fraction. These points are all raised in the abstract. We infer that the unexpected result not highlighted in the abstract is the observation that Sea Sweep-generated aerosol consistently had lower polysaccharide:Na+ ratios than ambient aerosol. We added this statement to the abstract:

"*Aerosol polysaccharide:sodium mass ratios were consistently higher in ambient air than in the artificially generated sea spray, and we hypothesize that this results from more rapid wet deposition of sodium-rich aerosol."*

The reviewer fairly critiques the interpretation that the data show a seasonal cycle in the relative abundance of polysaccharide on the basis that the number of sample points is small. This claim hinges on the 2-3 points available for each sample type in November. We have now performed two-sample t-tests to quantify the uncertainty about the seasonal differences and reported on them in the text:

"*Two-sided t-tests showed that both May and September polysaccharide:Na+ ratios were higher than in November at a significance level of 0.05.*"

"*Similarly, the potassium:sodium ratio was statistically significantly higher in September than in November at a significance level of 0.05*"

"The sub-180 nm aerosol showed some enhancement of recalcitrant factor with respect to sodium during September, particularly for ambient aerosol, *but the difference was not statistically significant at a 0.05 level.*"

*"Higher fractions, up to about 32%, were measured in late spring and summer in sub-180 nm aerosol for both Sea Sweep and ambient aerosol (different from November at a 0.05 significance level)…"*

Figure 8: I am concerned about the presentation of this data and its interpretation. The use of box and whisker plots to represent a dataset with a small number of samples may be leading to suspicious interpretations – or at least could confuse the interpretation of the data. What does a box and whisker plot mean when only three samples are included? The box clearly no longer represents an inter-quartile range… because how does one have quartiles with three samples? Or in panel (a), there is a group of 4 samples with a median value close to the lower end of the interquartile range, so is the 75th percentile and 90th percentile driven by just one point with a high ratio? It may be more useful to use a violin plot (showing the distribution of data rather than a box) or simply a plot of all of the data points without statistical treatment since the number of samples is so small. At the same time, while the authors clearly have a few data points for April, presenting the lack of data may confuse the interpretation. To be clear, it is understandable that only a small number of samples exist – the question is really about data reduction and associated interpretation.

We appreciate the reviewer's concern about the interpretation of these plots. The statistical tests described above should give more confidence about the differences that one infers from looking at the box-and-whisker plots. Similar to this format, a standard violin plot would also include interquartile measures that aren't calculable for the data with a small number of samples, so it is probably not a better choice. As an alternative, we simply plot the individual samples as points with a standard deviation for multiple analyses of the same sample in all the figures that previously had box-and-whisker plots. Now there can be no ambiguity about the statistical calculations and the data are simply there for the reader to see.

In addition, it would be helpful to the reader's interpretation of the data if some brief description of the conditions of the sampling location and/or general trends in biological productivity are described for each of the months shown. A reader who is less initiated with respect to the general seasonality or sampling plans for these North Atlantic cruises would benefit significantly from a short description of the conditions.

Agreed. We added as follows:

*"Four month-long scientific cruise deployments of the R/V Atlantis were carried out during NAAMES, and a thorough description of the scientific motivation and overview of the oceanographic and meteorological conditions can be found in Behrenfeld et al. (2019). The cruises were timed to sample critical periods of the annual cycle in phytoplankton: the transition to low phytoplankton concentrations in winter (N1, Nov. 2015), the early spring phytoplankton accumulation period (N4, March-April 2018), the spring peak period (N2, May 2016), and the summer "depletion phase" (N3, Sept. 2017). The main sampling periods occurred between ~39° and 54° N latitude, and stations during which Sea Sweep (described below) was deployed included cyclonic and anticyclonic eddies, as well as waters outside of eddies. As expected, the lowest chlorophyll levels were encountered in November and the highest in May."*

Lines 382 – 386: The early part of this set of lines is understandable with respect to the suggested association between CCN and sulfate. But then in the last line of the paragraph, polysaccharides are once again invoked. This transition back to polysaccharides is confusing. Is there evidence for polysaccharides to be in Na-free sea spray particles? Studies on this topic have been conducted. The question is not that this is speculative, but that the speculation is not well contextualized or supported by outside evidence.

Lines 398 - 400: See commentary about evidence for aerosol mixing state above. Sampling the literature on the mixing state of sea spray aerosol more thoroughly would improve this discussion substantially.

These are excellent points. We have added this statement after the speculation about polysaccharides in non-Na particles being CCN precursors:

*"Polysaccharide-like material has been identified in both sea salt-containing particles and sea salt-free marine particles (Hawkins and Russell, 2010). Smaller primary marine organic particles that may be important as*

*CCN precursors are particularly likely to be sea salt-free (Leck and Bigg, 2008)."*

*Hawkins, L. N. and Russell, L. M.: Polysaccharides, Proteins, and Phytoplankton Fragments: Four Chemically Distinct Types of Marine Primary Organic Aerosol Classified by Single Particle Spectromicroscopy, Adv. Meteorol., 2010, 1–14, doi:10.1155/2010/612132, 2010.*

*Leck, C. and Bigg, E. K.: Comparison of sources and nature of the tropical aerosol with the summer high Arctic aerosol, Tellus, Ser. B Chem. Phys. Meteorol., 60 B(1), 118–126, doi:10.1111/j.1600-0889.2007.00315.x, 2008.*

Minor comment Line 110: should be "mass spectrometric" as mass spectroscopy is not the proper name for mass spectrometry

Fixed.

Referee 2 Comments

The study by Lawler et al. is dedicated to applying state-of-the-art instrumentation to elucidate sea spray composition over the North Atlantic with a particular focus on transferrable water soluble organic species and their relationship to ocean biological activity. The main idea of the study to utilise TDCIMS is innovative and producing new insights into the subject. The main problem with the paper and forthcoming conclusions is the number of samples which makes it difficult in arriving at robust inferences due to inherent natural variability temporally as well as spatially. The authors did their very best in trying to squeeze as much information as possible from the data analysis and fairly interpreting their findings. However, due to scarcity of data the conclusions are rather vague and tend to overly rely on previous literature findings and seeking commonality, which is only natural given limited sample bank. I am not sure if the paper should rather be published in Atmospheric Measurement Techniques which would be a more suited medium for application of innovative techniques and somewhat limited dataset. ACP is instead reporting scientific advancements which are quite limited in this study due to overarching limitations. As an example of difficulties the authors confronted in terms of robustly derived conclusions is poor statistics. How can basic statistical variables be derived from e.g. 3 samples? Overall, the paper is written very well and follows clear structure, so apart from vague conclusions (not of authors fault) can possibly be published in ACP after addressing the comments.

We agree that a limitation of the paper is the relatively small number of samples, but we strongly disagree that only vague conclusions were reached or that the paper belongs in a technique-focused journal.

Perhaps this wasn't made sufficiently clear, but for several years, the large alcohol functional group contribution to primary marine aerosol found in multiple studies [e.g. Hawkins and Russell, 2010, Russell et al. 2010] was either not attributed to a source or was interpreted as generally representing polysaccharide material. This study shows strong evidence that they are not polysaccharides and that they are likely recalcitrant organics. We modified the following:

"We hypothesize that this factor represents recalcitrant dissolved organic matter (DOM) in seawater *and that by extension alcohol functional groups identified in marine aerosol may more typically represent recalcitrant DOM rather than biogenic saccharide-like material, contrary to inferences made in previous studies*."

Overall, we are reporting on 27 discrete day-long samples of marine conditions aerosol as well as 41 discrete station experiment samples in the open ocean. Marine organic aerosol composition data are relatively rare, and aerosol polysaccharide concentrations, let alone size-resolved, are all but nonexistent over most of the oceans. This is an unprecedented dataset that has yielded insights that will inform many future studies and provide context for past ones.

We agree that the ability to establish a seasonal cycle in the measured quantities is limited by the available data, in particular by the sparsity of samples from November. Reviewer 1 also noted this, and we have responded by providing two-sided t-test results that demonstrate statistically significant differences between November and the late spring/summer at a 0.05 significance level (see above). We also took Reviewer 1's

suggestion to plot the data as individual sample points to avoid inconsistencies in the box-and-whisker plots that make interpretation difficult.

Hawkins, L. N. and Russell, L. M.: Polysaccharides, Proteins, and Phytoplankton Fragments: Four Chemically Distinct Types of Marine Primary Organic Aerosol Classified by Single Particle Spectromicroscopy, Adv. Meteorol., 2010, 1–14, doi:10.1155/2010/612132, 2010.

Russell, L. M., Hawkins, L. N., Frossard, A. a, Quinn, P. K. and Bates, T. S.: Carbohydrate-like composition of submicron atmospheric particles and their production from ocean bubble bursting., Proc. Natl. Acad. Sci. U. S. A., 107(15), 6652–7, doi:10.1073/pnas.0908905107, 2010.

Comments in their sequence:

Title I think the title should be amended to "...thermal desorption mass spectroscopic approach:..." Later in the methods it is worded exactly that way.

As Reviewer 1 pointed out, "spectroscopic" is the wrong word here. We removed the word "approach": "…thermal desorption mass spectrometry: polysaccharides, …"

Line 89. It is unclear whether submicron fraction overlapped with <180 nm fraction or was it separate 180-1000nm size fraction?

They overlapped. We modified the text to clarify: *"Size cutoffs isolating <180 nm and <500 nm particles were made using independent single-stage Berner impactors, and a separate cyclone was used to isolate submicron particles."*

Line 95. The criteria are appropriate, but not very conservative. Typically, CN concentrations in clean marine atmosphere are <1000 cm-3, except for new particle production events. Radon concentration is typically more like <200mBq/m3 and BC<30ng/m3 (but depends on the measurement method and instrument). While the differences are not large, but it is important to observe that the increase in CN is not associated with the increase in BC or radon, which would immediately suggest diluted continental outflow or ship own exhaust. 48 hour trajectories are good, but should be checked if they stay in the boundary layer or were affected by large scale subsidence. In summary, I would argue for a more conservative and integrated approach, because typical marine back- ground is very low and can be easily perturbed by even by sporadic pollution events including own ship (stacks or various exhausts).

These criteria are supported by a thorough analysis of back trajectories and pollution thresholds over the NAAMES cruises (Saliba et al. 2020). That paper includes a comparison of the dataset with varying BC thresholds, largely showing that a <50 vs <25 ng/m3 does not make much of a difference for interpreting the results and of course allows more samples to be included in any analysis. It also includes cumulative probability distributions for BC and non-refractory organics and sulfate, separated by marine and continental periods, to show the effects of the reasonable but imperfect criteria used. The North Atlantic is not extremely clean much of the time, so here we are really distinguishing between "mostly marine" and "significant continental influence."

We added this statement:

"A thorough evaluation of these criteria and representative back trajectories can be found in Saliba et al. (2020)."

Saliba, G., Chen, C., Lewis, S., Russell, L. M., Quinn, P. K., Bates, T. S., Bell, T. G., Lawler, M. J., Saltzman, E. S., Sanchez, K. J., Moore, R., Shook, M., Rivellini, L., Lee, A., Baetge, N., Carlson, C. A. and Behrenfeld, M. J.: Seasonal Differences and Variability of Concentrations, Chemical Composition, and Cloud Condensation Nuclei of Marine Aerosol over the North Atlantic, J. Geophys. Res. Atmos.,, 0– 3, doi:10.1029/2020JD033145, 2020.

Line 110. How many samples a subset consisted of? It is not clear how many "clean marine" samples

were preferentially chosen. Later in the text and figures can be seen that probably 27 "clean marine" samples were analysed in two size fractions each. How many corresponded to Sea sweep and how many to the ambient? All that information should be stated right at the beginning. At the moment the sample set is completely under described.

We rewrote this paragraph to describe the dataset and included a table with sample numbers in the supplemental:

*"A subset of the ambient aerosol and sea sweep samples analyzed by FTIR were selected for mass spectrometric analysis by TDCIMS. An effort was made to analyze as many Sea Sweep samples as possible because we expected to see the strongest signals for primary organics and to be able to directly link them to seasonal phytoplankton changes in the underlying water. For ambient samples, periods characterized as marine were preferentially chosen. Periods of continental influence were included for comparison (13 samples). When possible, all size cuts in a given sample time period were analyzed by TDCIMS. Here we report only submicron and sub-180 nm because these were the two size fractions that were sampled for both ambient and Sea Sweep aerosol and can be directly compared. These amounted to a total of 41 Sea Sweep and 27 marine ambient samples reported (see Table S1), with roughly two-thirds of all Sea Sweep deployments represented with at least one sample and about 28% of marine-characterized ambient samples in these size classes."*

Line 149. NaCl fraction in sea salt is about 70%. What was the reason not choosing sea salt standard from Sigma-Aldrich or similar company? Na may respond differently in a more complex matrix in the instrument, but having sea salt matrix would certainly be compatible with sample extracts.

This is a fair point, but it doesn't impact the analyses here because we exclusively used the IC data for cation concentrations. We make that explicit here and add a plot in the supplemental to show the difference in response for the two methods for our samples.

*"The instrument response to $Na^+$ was characterized with NaCl (Sigma Aldrich) solutions in ultrapure water to compare with the ion chromatography results (below), but TDCIMS-derived $Na^+$ was not used for any subsequent analysis. The comparison is presented in Figure S6."*

Line 153. PMF value and purpose was developed for large datasets/matrixes. Having only a handful of samples what was the purpose of statistical multivariate technique, when it is well known that PMF output (and of every statistical technique) is more reliable with large number of samples? I believe rich spectroscopic signatures were the reason to analyse the results statistically, but still highly non-symmetrical matrix would hardly produce reliable results.

The clear distinctions among the factors in their mass spectra and in their different relationships to other measured aerosol quantities are evidence that the identified factors represent distinct organic aerosol types. To provide further assurance of the robustness of the PMF results, we performed bootstrapping experiments per instructions in the EPA PMF manual and report the results in the text:

*"A bootstrapping test was performed to assess the robustness of the 5 PMF factors when random subsets of the data were selected, using criteria recommended by the EPA PMF 5.0 software and manual (16-sample blocks, 100 bootstrap runs). The polysaccharide, fatty acid, and recalcitrant factors were matched on almost every run (100%, 100%, and 98% of the time respectively), indicating that these factors are distinguishable throughout the dataset. SOA 1 and 2 were matched for 83% and 69% of the runs, respectively. The sensitivity of the SOA factors to smaller sample subsets can be attributed to the strong influence of a few continental-influenced samples and to the fact that the Sea Sweep samples contain no or little secondary aerosol."*

Line 241. It is not stated that the correlations were statistically significant at whatever level which is especially important for a small dataset. Only then one can consider strong or modest relationships based on slopes or coefficient magnitude.

We added this statement to clarify the statistical significance of the relationships:

*"Since there were always at least 11 samples in the correlation analyses, this corresponds to at least a 0.005*

*significance level for strong relationships and at least a 0.05 significance level for modest relationships, based on a one-sided t-test."*

Line 264. It could well be a consequence of sea sweep device generating abnormal sea spray concentrations (Na) compared to wave breaking of plunging jets/sheets reminiscent of real waves. Correlating very large numbers with very small can disturb the pattern due to the variability of each.

We added this statement to address this point:

*"This could result from a greater, more highly variable contribution of larger, salt-dominated particles in the Sea Sweep samples."*

Line 385. What does internally mixed NaCl represent? NaCl cannot be considered separately from other sea salt ions all of which simultaneously make sea salt. Sea salt can be internally mixed with organics and if organics is primary then both are termed sea spray.

We revised this statement:

''Sea spray aerosol was only found to correlate with CCN in the winter during NAAMES, *but in that study sea spray was treated as a single internally mixed mode."*

Line 396. Coagulation is extremely unlikely given very low particle numbers in clean marine air. Textbook coagulation requires many orders of magnitude higher aerosol concentrations. Sea sweep is probably different in relation to number concentration, but in any event sea sweep is poor representation of real world wave breaking produced sea spray.

We are referring to coagulation in seawater. We think this is already sufficiently clear in the text but added references:

"This is perhaps not surprising, given that the recalcitrant organic material in seawater is dissolved and well-mixed *(Hansell 2013)*, while the polysaccharides are likely particulate and/or colloidal and heterogeneously distributed due to their bioavailability and tendency to coagulate *(Wurl et al. 2011)*."

*Hansell, D. A.: Recalcitrant Dissolved Organic Carbon Fractions, Ann. Rev. Mar. Sci., 5(1), 421–445, doi:10.1146/annurev-marine-120710-100757, 2013.*

*Wurl, O., Miller, L. and Vagle, S.: Production and fate of transparent exopolymer particles in the ocean, J. Geophys. Res. Ocean., 116(12), 1–16, doi:10.1029/2011JC007342, 2011.*

Line 398. All sea spray aerosol is born as a sea water droplets. Organic matter species cannot be taken out of sea water in a dry state. Not even concentrated microlayer film can be without some amount of sea water. Therefore, it is inevitable that at least a small amount of sea salt would be present in an aerosol particle even if organic species would overwhelmingly dominate aerosol particle volume. As a consequence, primary sea spray components cannot be considered externally mixed, because they are variants of internal mixture. Only secondary components can be externally mixed, because they originate via different process (gas-to-particle conversion).

This distinction may be a matter of degree. For example, if 20 nm sea spray particles consist of a mixture of particles that are 99% organic-1% salt and 99% salt-1% organic, we consider it reasonable to describe that as an external mixture for the purposes of characterizing aerosol populations. Clearly, primary marine particles that are almost entirely organic exist (e.g. Hawkins and Russell 2010). Other authors have also described primary sea spray with different compositions arising from film vs. jet droplets as an external mixture (e.g. Wang et al. 2017).

Hawkins, L. N. and Russell, L. M.: Polysaccharides, Proteins, and Phytoplankton Fragments: Four Chemically Distinct Types of Marine Primary Organic Aerosol Classified by Single Particle Spectromicroscopy, Adv. Meteorol., 2010, 1–14, doi:10.1155/2010/612132, 2010.

Wang, X., Deane, G. B., Moore, K. A., Ryder, O. S., Stokes, M. D., Beall, C. M., Collins, D. B., Santander, M. V., Burrows, S. M., Sultana, C. M. and Prather, K. A.: The role of jet and film drops in controlling the mixing

state of submicron sea spray aerosol particles, Proc. Natl. Acad. Sci., 114(27), 6978–6983, doi:10.1073/pnas.1702420114, 2017.

Figure 4. How can meaningful statistics be extracted having 3-5 samples in each type? Even 7 samples is barely enough. Presenting average and a range is all can be done here. September submicron samples suggest negative polysaccharide concentration - what does that mean?

We have replaced the box plots with the individual sample points here and in Figures 8 and 9, so no per-season statistical variables need to be calculated anymore. A method blank was subtracted from the polysaccharide samples, as stated in methods. This can lead to negative background-corrected data in the case of small samples. We modified section 3.2.1:

"A handling background of 0.15 µg for each filter was subtracted on the basis of the field blank concentrations. *This had little effect on the ratio analysis described below but resulted in a small number of negative polysaccharide concentrations. These are plotted to show method precision in absolute concentration plots and set to 0 for ratio plots. The estimated detection limit was about 0.12 µg polysaccharide material on a filter, or roughly 0.01 µg m$^{-3}$ for ambient air samples on the basis of calibration standards.*"

And this statement in the caption for the relevant figure:

"*One sample was lower than the estimated background, resulting in a negative calculated concentration.*"

---

## Author Response (AR2)

Author responses in blue following editor comments. Changes to the manuscript are italicized.

Editor Comments:

I would like to thank the authors for addressing all the comments raised by the reviewers. I have only a few comments.

We thank the editor for the timely review of our manuscript.

Concerning figure 4, the reviewer pointed out the presence of a few number of negative data-points and the authors explained the result as the effect of blank subtraction. I would like to ask the authors to verify if the negative data-point in September had a concentration above detection limit before blank subtraction.

We added this statement to the caption of Figure 4:
*"All samples were above the estimated detection limit prior to background subtraction."*

In addition, evaluate the possibility to merge Table 1 with Table 2 and figures 3,5,6,7 in one single figure.

We merged the specified figures into a single figure and find that this makes the comparison between the PMF factors easier for the reader to see.

We appreciate the idea that merging Tables 1 and 2 would bring the ambient and Sea Sweep data right next to each other, but we find that the added complexity makes a merged table significantly more difficult to interpret.

[revised manuscript text omitted]